# Radiology Reports Improve Visual Representations Learned from Radiographs

**Haoxu Huang**                                                                 HH2740@NYU.EDU
*Courant Institute of Mathematical Sciences, New York University, USA*
**Samyak Rawlekar**                                                             SKR2369@NYU.EDU
*Tandon School of Engineering, New York University, USA*
**Sumit Chopra**                                          SUMIT.CHOPRA@NYULANGONE.ORG
**Cem M. Deniz**                                              CEM.DENIZ@NYULANGONE.ORG
*Department of Radiology, New York University Langone Health, USA*

**Editors:** Accepted for publication at MIDL 2023

## Abstract

Although human's ability to visually understand the structure of the World plays a crucial role in perceiving the World and making appropriate decisions, human perception does not solely rely on vision but amalgamates the information from acoustic, verbal, and visual stimuli. An active area of research has been revolving around designing an efficient framework that adapts to multiple modalities and ideally improves the performance of existing tasks. While numerous frameworks have proved effective on natural datasets like ImageNet, a limited number of studies have been carried out in the biomedical domain. In this work, we extend the available frameworks for natural data to biomedical data by leveraging the abundant, unstructured multi-modal data available as radiology images and reports. We attempt to answer the question, "For multi-modal learning, self-supervised learning and joint learning using both learning strategies, which one improves the visual representation for downstream chest radiographs classification tasks the most?". Our experiments indicated that in limited labeled data settings with 1% and 10% labeled data, the joint learning with multi-modal and self-supervised models outperforms self-supervised learning and is at par with multi-modal learning. Additionally, we found that multi-modal learning is generally more robust on out-of-distribution datasets. The code is publicly available online [1]

**Keywords:** Multi-Modal Learning, Self-Supervised Learning, Out-of-Distribution, Radiology

## 1. Introduction

Multi-modal learning(Xue et al., 2018; Miura et al., 2020; Monshi et al., 2020) has become an extremely popular area of research in deep learning. While a good amount of multi-modal learning research has been conducted on datasets like ImageNet (Deng et al., 2009) which consists of natural images (Radford et al., 2021; Kim et al., 2021; Bardes et al., 2022b; He et al., 2020; Chen et al., 2020a; Grill et al., 2020; Zbontar et al., 2021; Bao et al., 2022b; Li et al., 2021b; Bardes et al., 2022a; Mu et al., 2022; Caron et al., 2020), a limited number of research have been conducted in the biomedical domain (Boecking et al., 2022; Tiu et al., 2022; Zhang et al., 2020; Chaitanya et al., 2020; Haghighi et al., 2022; Azizi et al.,

---

1. Code: https://github.com/denizlab/MIMICCXR-MultiModal-SelfSupervision.

2021; Seibold et al., 2022). In biomedical domain, the choice between multi-modal and self-supervised learning is still unclear for learning better visual representations. Although few studies similar to ConVIRT (Zhang et al., 2020) have looked into the quality of visual representation for biomedical images on multi-modal vs. self-supervised learning, their evaluations are conducted only on a small fraction of dataset for self-supervised learning on a part of most important pathology, which is not guaranteed to see which methods generally perform better. Hence, we believe it is important to compare the performances of multi-modal, self-supervised learning and their joint training in a more comprehensive way.

Biomedical data can effortlessly take advantage of multi-modal learning. For example, there is a paired report for a given radiograph, and MRI scans generally contain multiple contrasts and reports. While many multi-modal models can be used off-the-shelf (Jaegle et al., 2021; Mu et al., 2022; Radford et al., 2021), they are usually trained on natural image datasets which are ten orders of magnitude larger than the available biomedical datasets. While several groups have conducted experiments on biomedical multi-modal learning (Tiu et al., 2022; Boecking et al., 2022; Seibold et al., 2022), their main focus was limited on evaluating the model performance with multi-modal input. To our knowledge, comparative research has yet to be conducted among multi-modal, self-supervised, and joint training on the out-of-distribution (OOD) chest radiograph classification tasks. The aim is to identify the optimal approaches for large scale unlabelled multi-modal radiograph datasets. Our contributions are summarized as follows:

- For OOD chest radiographs, we observe that multi-modal learning generally provides higher quality visual representation.

- Under limited supervised training data, we empirically identify multi-modal and joint training have better performance than self-supervised for chest radiographs. Under large scale supervised training data, the performance of self-supervised learning is at par with multi-modal learning and joint training.

- Provided a benchmark for contrastive learning-based self-supervised, multi-modal learning and their joint training for chest radiographs with code available for further investigation.

## 2. Related Works

**Multi-modal learning** (Radford et al., 2021; Zhang et al., 2020; Li et al., 2021a, 2022; Yu et al., 2022) learn representations for both image and text data concurrently. One of the most popular approaches of multi-modal learning is learning from image and text data contrastively, where the representation for paired image and text are aligned closer and unpaired image and text are dispelled further. Approaches like CLIP (Radford et al., 2021) and ConVIRT (Zhang et al., 2020) are considered be the representative work of contrastive multi-modal learning for natural and medical images, respectively.

**Self-supervised learning** is a recent popular topic in representation learning, where the model learns the patterns by training on unlabeled data. Three different self-supervised learning approaches, SimCLR (Chen et al., 2020a), MoCoV2 (Chen et al., 2020b), and

VICReg (Bardes et al., 2022b) were evaluated in our paper. The general idea of the self-supervised learning methods we evaluated is that they train the model to learn the invariant representation of the same image by applying different augmentations. Although Masked Image Modeling based self-supervised learning such as MAE (He et al., 2022), BEiT (Bao et al., 2022a), etc. become popular in recent years, we did not evaluate their performance in this study because their objective is not consistent with the idea of enforcing invariant representation for the self-supervised and multi-modal learning we experimented with.

**Application in biomedical domain:** CheXzero (Tiu et al., 2022) explored the performance of CLIP on radiology chest scan diagnosis by taking in language and vision input together with zero-shot prompting. BioVLP (Boecking et al., 2022) explored the performance of contrastive self-supervised learning with masked language modeling on radiology chest scan diagnosis. ConVIRT (Zhang et al., 2020) explored if multi-modal learning on radiology image-text pair can learn strong image representation. GLoRIA (Huang et al., 2021) shows adding local representation during multi-modal learning on radiology image-text pair can improve performance upon ConVIRT. (Azizi et al., 2021) shows image representations learned from the medical images by SimCLR can improve downstream performance. REMEDIS (Azizi et al., 2022) performed a large-scale evaluation on both in-distribution (ID) and OOD medical images with SimCLR.

## 3. Methods

The diagram of the experimental process is shown in Figure 1. For pre-training, we train the model with multi-modal learning, self-supervised learning or joint training by optimizing over both multi-modal learning and self-supervised learning as proposed by (Mu et al., 2022). For evaluating the pre-trained model quality, we finetune the pre-trained model with additional classification heads. Specifically, we performed linear probing (LP), fine-tuning (FT) or fine-tuning after linear probing (LPFT) (Kumar et al., 2022).

### 3.1. Self-Supervised and Multi-Modal Pretraining

We used the following pretraining strategies for the comparative study: we chose CLIP (Radford et al., 2021) and ConVIRT (Zhang et al., 2020) as they are two earliest representative works on contrastive multi-modal learning. We chose three most representative categories for self-supervised learning that enforcing invariant of representations as SimCLR (Chen et al., 2020a) (traditional contrastive learning), VICReg (Bardes et al., 2022b) (redundancy reduction) and MoCoV2 (Chen et al., 2020b) (caching negative representations with dictionary for contrastive learning).

**CLIP (Radford et al., 2021):** Given image representation $u$ and text representation $v$, CLIP loss is formulated as

$$\mathcal{L} = \mathcal{L}_{i2t} + \mathcal{L}_{t2i} \tag{1}$$

where

$$\mathcal{L}_{i2t} = -\sum_{i\in\mathcal{B}} \log \frac{\exp(\tau u_i v_i)}{\sum_{j\in\mathcal{B}} \exp(\tau u_i v_j)}, \ \mathcal{L}_{t2i} = -\sum_{j\in\mathcal{B}} \log \frac{\exp(\tau u_i v_i)}{\sum_{i\in\mathcal{B}} \exp(\tau u_i v_j)} \tag{2}$$

where $\mathcal{B}$ is batch size and $\tau$ is temperature hyperparameter .

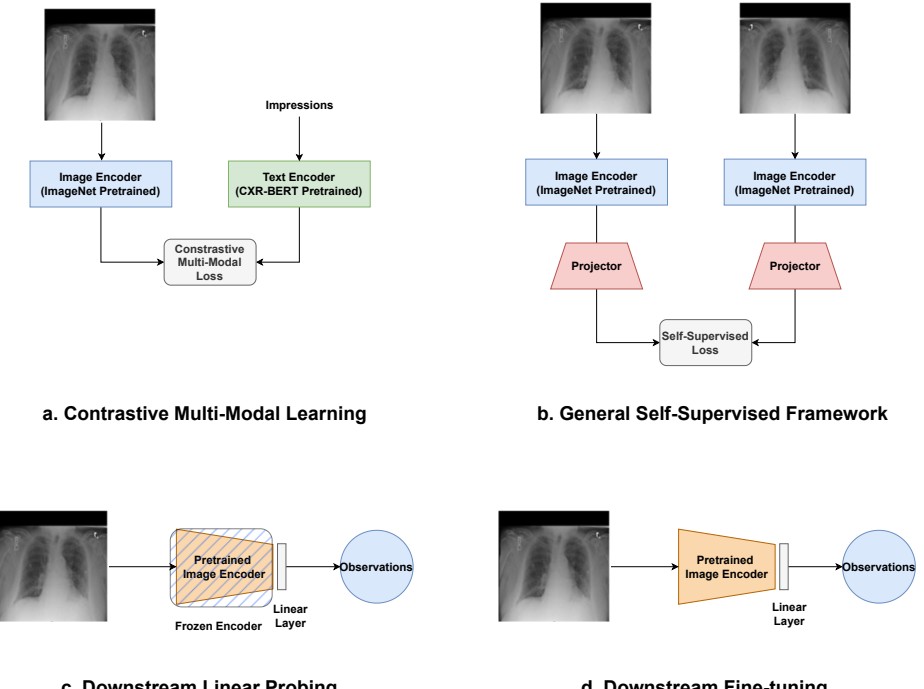

Figure 1: Experimental Process: a) Contrastive multi-modal learning training. b) Contrastive self-supervised learning training. c) Downstream linear probing (**LP**). d) Downstream fine-tuning (**FT**).

**ConVIRT (Zhang et al., 2020):** ConVIRT followed the same loss as CLIP, but it is more fine-grained toward medical imaging with more specific augmentations (we adopted these augmentations as our general multi-modal learning augmentation mentioned in Appendix B.3) and architecture design choices that freeze top six layers of text encoder and use shallower projection heads.

**SimCLR (Chen et al., 2020a):** Given two input representations $z_i$ and $z_j$ in a given batch. The NT-Xent loss by SimCLR is formulate as

$$l_{i,j} = -\log \frac{\exp(sim(z_i, z_j)/\tau)}{\sum_{k=1}^{2N} \mathbb{1}_{k \neq i} \exp(sim(z_i, z_k)/\tau)} \tag{3}$$

where $\tau$ is temperature hyperparameter, $N$ is number of in batch samples, $sim$ is the similarity metric set to be a dot product following the original implementation.

**MoCoV2 (Chen et al., 2020b):** We followed the original proposal of MoCo that creates a very large memory bank to cache image representations with InfoNCE (van den Oord et al., 2018) in order to introduce more negative samples within a single batch. The InfoNCE loss is formulated as

$$\mathcal{L} = -\log \frac{\exp(q \cdot k_+/\tau)}{\sum_{i=0}^{K} \exp(q \cdot k_i/\tau)} \tag{4}$$

where $q$ is the representation for anchor input, $k_+$ is the representation for positive input, $k_i$ is the representation for all cached input in memory bank and $\tau$ is a temperature parameter. The sample representations in the memory bank is gradually updated by a separate encoder that is update by trained encoder with Exponential Moving Average (EMA).

**VICReg (Bardes et al., 2022b):** Given two input representations $Z$ and $Z'$ obtained by passing two transformed views of same image, VICReg loss is computed as

$$\ell(Z, Z') = \lambda s(Z, Z') + \mu[v(Z) + v(Z')] + \nu[c(Z) + c(Z')] \tag{5}$$

where $s(Z, Z') = \frac{1}{n} \sum_i \|z_i - z'_i\|_2^2$ is the invariance term for calculating the feature-wised mean-squared euclidean distance with given feature size $n$. $v(Z) = \frac{1}{d} \sum_{j=-1}^{d} \max(0, \gamma - S(z^j, \epsilon))$ is the variance term for given number of samples $d$, $S(x, \epsilon) = \sqrt{\mathrm{Var}(x) + \epsilon}$ is the regularized standard deviation with $\gamma$ as a constant target standard deviation value, $\epsilon$ as a scalar to enforce numerical stability. $C(Z) = \frac{1}{n-1} \sum_{i=1}^{n}(z_i - \bar{z})(z_i - \bar{z})^T$ is the covariance term where $\bar{z} = \frac{1}{n} z_i$. $\lambda$, $\mu$, $\nu$ are hyperparameters to control the contribution of each loss term.

**Joint Training:** The joint training of multi-modal and self-supervised learning was proposed by SLIP (Mu et al., 2022) for natural image training. Given image and text representations $u$ and $v$ from CLIP projector, and two projected augmented image representations from some self-supervised learning projector $z_i$ and $z_j$, SLIP is trained by minimizing

$$\ell_{slip} = c \cdot \ell_{ssl}(z_i, z_j) + \ell_{clip}(u, v) \tag{6}$$

where $\ell_{ssl}$ is the loss for some self-supervised method and $c$ is a hyperparameter to control the contribution of self-supervised loss. In our experiments, we chose multi-modal learning methods to be CLIP and ConVIRT, self-supervised learning methods to be SimCLR, MoCoV2 and VICReg. While SimCLR and MoCoV2 with CLIP was experimented in the original SLIP paper, VICReg and ConVIRT were not experimented in the original work.

## 4. Experimental Setup

See Appendix B for details on hyperparameters for both pre-training and downstream training.

### 4.1. Dataset

Our multi-modal frameworks were trained on MIMIC-CXR (Johnson et al., 2019) dataset, which consists of 377,110 pairs of chest X-ray image-report pairs collected from Beth Israel Deaconess Medical Center in Boston, MA, USA. Radiology reports, although an excellent source of information, carry redundant information. Findings and impressions are richer in content. Findings present the diagnosis from the given radiology image, and impressions presents the complete narrative description from doctors according to multiple radiology images of the same patient. In our experiment, we only chose impressions as our text input because impressions have more concise description of the patients diagnosis, where we found it introduced less noise than findings.

Our trained models were evaluated on CheXpert (Irvin et al., 2019) and NIH-ChestX-ray14 (Wang et al., 2017) dataset with LP on 100% data, LPFT on 100% data and FT

on 1%, 10% and 100% data (Kumar et al., 2022). CheXpert dataset consists of 224,316 chest radiographs of 65,240 patients from Stanford Hospital in Palo Alto, CA, USA. There are 14 observed pathologies labeled by the radiologist as positive, negative, or uncertain. The classification here is multi-class(14 pathologies) and multi-label(each image can belong to multiple pathalogies). For our experiments, we followed the train-validation (224,316-200) split used in official paper. Our final results are on the validation dataset as the test set is hidden for leader-board. The NIH-ChestX-ray14 (Wang et al., 2017) dataset consists of 112,120 frontal-view X-ray images of 30,805 collected at the National Institutes of Health Clinical Center, MD, USA, with unique patients and the text-mined fourteen common disease labels. This datasets is also multi-class and multi-label. We followed the train-validation-test split (70%-10%-20%).

The medical domain also faces the problem of OOD, in which even though the input data from two different datasets is chest radiographs, however, the origin, extraction methods and target population affect the distribution. In our case, both CheXpert(Irvin et al., 2019) and NIH-ChestXray 14(Wang et al., 2017) are OOD with MIMIC-CXR (Johnson et al., 2019), which we used for pre-training, because they are collected from very different regions.

### 4.2. Backbone Encoders

For all frameworks, we fixed the image encoder to be ResNet-50 (He et al., 2016). For the text encoder, we used BERT-base (Devlin et al., 2019). The image encoder was initialized with ImageNet pre-trained weights, and the text encoder was initialized with CXR-BERT-general pre-trained weights (Boecking et al., 2022).

### 4.3. Data Preprocessing

We loaded MIMIC-CXR images from corresponding DICOM files, and the images were resized to $224 \times 224$. Only impressions were used as text input. Samples with missing impressions were removed from training. For data augmentation, we used random resized crop, random horizontal flip, color jittering, gaussian blur, and solarization for self-supervised learning methods. In addition to the augmentations for self-supervised methods, for multi-modal approaches we used random affine transform. ImageNet normalization was used in all experiments.

## 5. Results

We show our benchmarking result in Table 1. All results were evaluated with the mean AUROC from all pathology. Mean AUROC: we first calculate AUROC for each individual pathology and then take the average over the AUROC of 14 pathologies for each of the datasets. We further explain them in section 5.1. We picked one method from multi-modal learning (CLIP), self-supervised learning (SimCLR) and joint training (SimCLR-CLIP) to analyze per pathology performance shown in Appendix D. Generally, we observed that using multi-modal learning or joint training improved the performance of pathologies that have low prevalence and the trend is more prominent with limited data. We further visualize some of our classification results with GradCam (Selvaraju et al., 2017) in Appendix C. In the original SimCLR paper, a significant performance improvement was observed as the

|     | Methods | CheXpert (AUROC) | | | | |
| --- | --- | --- | --- | --- | --- | --- |
|     |     | LPFT | LP | FT (1%) | FT (10%) | FT (100%) |
|     | Random Init. | $77.35 \pm 0.61$ | $62.91 \pm 0.62$ | $62.88 \pm 1.13$ | $70.36 \pm 0.98$ | $78.28 \pm 1.10$ |
|     | ImageNet Init. | $83.67 \pm 0.88$ | $67.12 \pm 0.24$ | $69.33 \pm 0.15$ | $77.98 \pm 0.28$ | $82.29 \pm 2.79$ |
|     | CLIP | $\mathbf{85.06} \pm 0.85$ | $78.17 \pm 0.23$ | $\mathbf{78.19} \pm 1.80$ | $82.90 \pm 1.46$ | $84.36 \pm 3.07$ |
|     | ConVIRT | $84.34 \pm 0.99$ | $\mathbf{79.92} \pm 0.11$ | $76.96 \pm 2.34$ | $82.89 \pm 2.40$ | $85.20 \pm 1.44$ |
|     | MoCoV2 | $81.95 \pm 1.78$ | $71.37 \pm 0.17$ | $74.33 \pm 1.53$ | $79.58 \pm 1.30$ | $84.12 \pm 1.08$ |
| $(a)$ | SimCLR | $84.24 \pm 0.95$ | $68.70 \pm 0.39$ | $72.83 \pm 0.91$ | $78.33 \pm 0.88$ | $84.74 \pm 0.96$ |
|     | VICReg | $82.81 \pm 0.67$ | $71.05 \pm 0.48$ | $72.16 \pm 1.83$ | $78.51 \pm 1.18$ | $\mathbf{85.40} \pm 1.63$ |
|     | MoCoV2-CLIP | $83.43 \pm 2.21$ | $76.89 \pm 0.16$ | $77.61 \pm 1.57$ | $80.29 \pm 0.71$ | $82.92 \pm 1.24$ |
|     | MoCoV2-ConVIRT | $84.81 \pm 1.82$ | $77.02 \pm 0.14$ | $77.93 \pm 1.43$ | $\mathbf{83.45} \pm 1.79$ | $83.51 \pm 1.37$ |
|     | SimCLR-CLIP | $85.10 \pm 0.72$ | $78.02 \pm 0.13$ | $77.84 \pm 0.99$ | $81.67 \pm 2.76$ | $84.33 \pm 0.79$ |
|     | SimCLR-ConVIRT | $83.95 \pm 1.66$ | $75.97 \pm 0.10$ | $76.68 \pm 1.61$ | $82.57 \pm 1.55$ | $84.39 \pm 1.09$ |
|     | VICReg-CLIP | $82.72 \pm 1.71$ | $69.01 \pm 0.27$ | $75.06 \pm 1.64$ | $81.17 \pm 1.61$ | $82.11 \pm 1.34$ |
|     | VICReg-ConVIRT | $83.05 \pm 1.25$ | $71.72 \pm 0.36$ | $74.93 \pm 2.27$ | $81.17 \pm 2.34$ | $83.05 \pm 1.76$ |
|     | Methods | NIH-ChestX-Ray14 (AUROC) | | | | |
|     |     | LPFT | LP | FT (1%) | FT (10%) | FT (100%) |
|     | Random Init. | $78.97 \pm 0.13$ | $62.54 \pm 0.34$ | $58.30 \pm 2.06$ | $70.06 \pm 1.02$ | $78.52 \pm 0.17$ |
|     | ImageNet Init. | $82.96 \pm 0.12$ | $75.46 \pm 0.78$ | $67.18 \pm 0.07$ | $76.44 \pm 0.63$ | $82.70 \pm 0.23$ |
|     | CLIP | $\mathbf{83.84} \pm 0.19$ | $\mathbf{83.80} \pm 0.16$ | $\mathbf{72.95} \pm 1.31$ | $\mathbf{79.80} \pm 0.67$ | $83.74 \pm 0.22$ |
|     | ConVIRT | $83.80 \pm 0.05$ | $83.48 \pm 0.04$ | $71.20 \pm 0.98$ | $79.24 \pm 0.89$ | $\mathbf{83.92} \pm 0.13$ |
|     | MoCoV2 | $83.20 \pm 0.18$ | $77.90 \pm 0.31$ | $68.50 \pm 0.99$ | $77.22 \pm 0.67$ | $83.10 \pm 0.37$ |
| $(b)$ | SimCLR | $82.82 \pm 0.19$ | $77.00 \pm 0.10$ | $69.01 \pm 0.41$ | $76.52 \pm 0.56$ | $82.76 \pm 0.32$ |
|     | VICReg | $82.80 \pm 0.26$ | $75.14 \pm 0.65$ | $69.06 \pm 0.47$ | $75.98 \pm 0.87$ | $82.74 \pm 0.24$ |
|     | MoCoV2-CLIP | $83.68 \pm 0.12$ | $81.42 \pm 0.04$ | $71.28 \pm 0.57$ | $79.02 \pm 0.52$ | $83.32 \pm 0.46$ |
|     | MoCoV2-ConVIRT | $83.62 \pm 0.40$ | $81.00 \pm 0.02$ | $71.10 \pm 0.81$ | $78.58 \pm 0.70$ | $83.62 \pm 0.13$ |
|     | SimCLR-CLIP | $83.80 \pm 0.12$ | $80.98 \pm 0.09$ | $71.33 \pm 0.99$ | $79.30 \pm 0.69$ | $83.66 \pm 0.18$ |
|     | SimCLR-ConVIRT | $83.38 \pm 0.10$ | $80.40 \pm 0.09$ | $70.90 \pm 0.67$ | $78.52 \pm 0.67$ | $83.45 \pm 0.08$ |
|     | VICReg-CLIP | $83.30 \pm 0.18$ | $78.02 \pm 0.11$ | $70.76 \pm 0.94$ | $78.22 \pm 0.60$ | $83.32 \pm 0.34$ |
|     | VICReg-ConVIRT | $83.24 \pm 0.20$ | $76.92 \pm 0.04$ | $70.80 \pm 0.76$ | $78.38 \pm 0.90$ | $83.20 \pm 0.25$ |

Table 1: We report the 95% confidence interval (CI) obtained from 5 different seeds in the form of (Mean $\pm$ CI range ) for the AUROC score of the chosen datasets with fine-tuning after linear probing (**LPFT**), linear probing (**LP**) (LPFT and LP are both on **100%** data), **1%**, **10%** and **100%** of labeled data fine-tuning (**FT**)

batch size increased. To further investigate this, we conducted additional experiments with SimCLR using batch sizes of 1024 and 2048. These results were then compared to those obtained from CLIP models trained with the same batch sizes, as shown in Appendix E. However, our findings reveal a consistent trend with our primary experiments.

## 5.1. Downstream Evaluation

For both OOD datasets we evaluated on, we observed that either FT with 100% data or LPFT has a comparable performance on self-supervised, multi-modal, and joint training. For FT with 1% and 10% data, self-supervised methods decay on performance fast, but multi-modal and joint training have much slower decay. For LP, we observed the same trend as fine-tuning on limited data: multi-modal and joint training perform much better than self-supervised learning. Empirically, we observed that results from NIH-ChestX-Ray with LP was very close to 100% FT and LPFT, which signifies that the two datasets

have a very similar distribution compared to CheXpert dataset. Another trend that we observed was introducing the additional information from radiology report helped the model to learn better visual representation than directly performing self-supervised learning. This phenomenon can be attributed to radiology reports, because they are very precise, clear and in a consistent format. Thus they contain highly semantic and dense information which complements the visual representation from images. This explanation is also identified by (Santurkar et al., 2023), where they found that the descriptiveness and variability (a.k.a style consistency among the text data) directly determine how well the models pre-trained by multimodal learning transfer learned information.

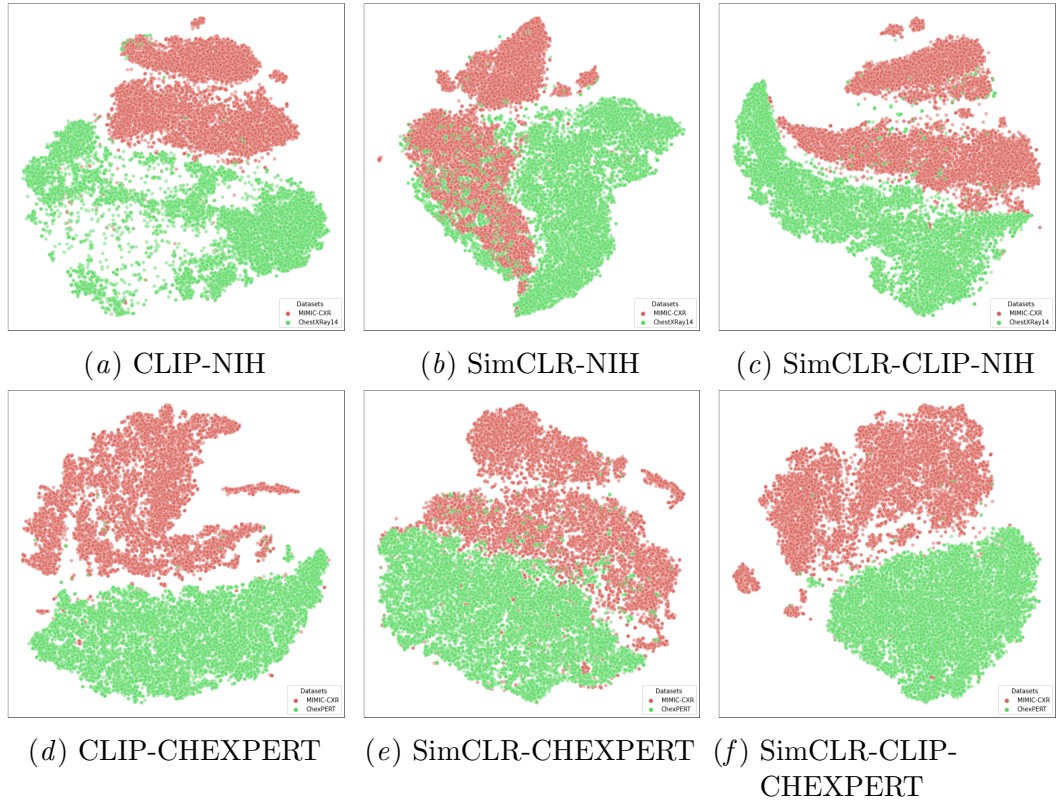

$(a)$ CLIP-NIH $\qquad$ $(b)$ SimCLR-NIH $\qquad$ $(c)$ SimCLR-CLIP-NIH

$(d)$ CLIP-CHEXPERT $\quad$ $(e)$ SimCLR-CHEXPERT $\;$ $(f)$ SimCLR-CLIP-CHEXPERT

Figure 2: Figures for t-SNE embedding with LPFT downstream training strategies with CLIP, SimCLR, SimCLR-CLIP pretrained backbone

## 5.2. t-SNE visualization

To identify the extent of OOD among the chosen datasets. We visualized the embedding of the backbone by t-SNE (van der Maaten and Hinton, 2008). For this experiment, we randomly picked $10,000$ samples of each dataset and plotting the output of the backbone trained with LPFT. The visualizations are shown in Figure 2. In the figure, the first row is trained on the NIH dataset, so the model learns to classify/identify the elements of the NIH. And with the addition of MIMIC dataset it is for classification into two classes. We can see

a visible boundary between MIMIC and NIH. Which justifies that the two datasets are from different distributions. If they were from the same distribution the learned backbone would not be able to create two clusters. Similar is the case for the second row with CheXpert data. Another observation is less separability when using self-supervised methods and a clear separation with multi-modal and joint training. This finding is coherent with our previous statement that multi-modal learning and joint training generate better representations. Additional visualizations in Appendix F shows more visualization on MoCoV2.

## 6. Conclusion

In this work, we observed that multi-modal learning and joint training performs better than self-supervised learning with limited supervised data, but the performance difference among all methods is largely decreased when data size scales up. In addition, classes that have fewer labels in the training data tend to experience greater benefits from multi-modal learning when applied to downstream tasks. For NIH-ChestX-ray 14 datasets, we found that initial representation learned by multi-modal learning and joint training is very strong with LP to be getting very close performance as 100% FT which implies that the two datasets have more similar distribution compared to CheXpert dataset. We hope our work would provide a good reference for future research on both multi-modal and self-supervised learning for medical imaging.

## Acknowledgments

Research reported in this publication was supported in part by the Center for Advanced Imaging Innovation and Research (CAI2R), a National Center for Biomedical Imaging and Bioengineering operated by NYU Langone Health and funded by the National Institute of Biomedical Imaging and Bioengineering through award number P41EB017183. This content is solely the responsibility of the authors and does not necessarily represent the official views of the National Institutes of Health.

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

## Appendix A. Computational Cost

We trained all models on four NVIDIA Tesla V100 GPUs with 32GB RAM for MIMIC-CXR pre-training. For all downstream experiments on CheXpert and NIH-ChestXRay-14 dataset, we used one NVIDIA Tesla V100 GPUs with 16GB RAM. The pre-training took roughly 1.5 to 2 days and fine-tuning took roughly 0.5 days.

## Appendix B. Hyperparameters Details

### B.1. Pre-Training

For pre-training, we trained all methods with a learning rate $1e$-4, batch size 256 with ResNet-50 backbone. We used AdamW optimizer (Loshchilov and Hutter, 2019) with cosine scheduling as optimizer scheduler (Loshchilov and Hutter, 2016) without warmup steps, beta2 of AdamW was set to be 0.98 for more stable training. The total number of epochs was set to 50 for training. Early stopping was used based on the loss on validation set. Weight Decay was set to be $1e$-5 for self-supervised learning and 0.01 for multi-modal learning. For fairness of comparisons, all self-supervised learning and multi-modal learning used the same augmentations as mentioned in Section 4.3. We did not perform any augmentation on text data.

### B.2. Downstream Training

For downstream training, we used $1e$-4 learning rate and Adam optimizer for both datasets. We set the batch size to 64 for CheXpert and 16 for NIH-Chest-Xray14. Random Horizontal Flip and Center Crop were performed on CheXpert and no augmentation was applied to NIH-Chest-Xray14. We trained CheXpert for 5 epochs and picked the epoch with highest AUROC. We trained NIH-Chest-Xray14 for 30 epochs and picked the epoch based on validation loss. For LPFT, we trained upon best epoch on LP with additional 5 or 30 epochs and picked the best epoch after FT based on corresponding dataset. We set temperature for loss function as 0.07 for CLIP and MoCoV2, 0.1 for ConVIRT and SimCLR. We set the self-supervised loss contribution weight hyperparameter $c$ in joint training to be 1.0.

### B.3. Data Augmentation Details

For multi-modal learning, we performed Random Resized Crop with size 224 and scale from 0.6 to 1.0, Random Horizontal Flip with probability 0.5, Random Affine with degree $-20$ to 20, translation 0.09 to 0.10 and scale 0.95 to 1.05, Color Jitter with brightness 0.6 to 1.4 and contrast 0.6 to 1.4 with probability 0.5, Gaussian Blur with min 0.1, max 3.0 and probability 0.5.

For self-supervised learning, we performed Random Resized Crop with size 224 and scale from 0.75 to 1.0, Random Horizontal Flip with probability 0.5, Color Jitter with brightness 0.4 contrast 0.4 and probability 0.8, Gaussian Blur with probability either 1.0 or 0.1 with min 0.1 and max 2.0. Solarization with probability either 0.2 or 0.

### B.4. Feature Projection Details

The image embeddings were originally projected to 2048 dimension and then further projected to 512 dimension for calculating CLIP loss. The text embeddings were originally extracted from last layer of $[CLS]$ token and then projected to 512 dimension as well. For SimCLR and MoCoV2, the image representations were re-projected to 512 dimension and then projected to 128 dimensions. For VICReg, the image representations were re-projected to 8192 dimensions with three linear layers that all have 8192 dimensions. We used batch normalization (Ioffe and Szegedy, 2015) for all projectors as in-between layers.

## Appendix C. Sample Classification Visualization

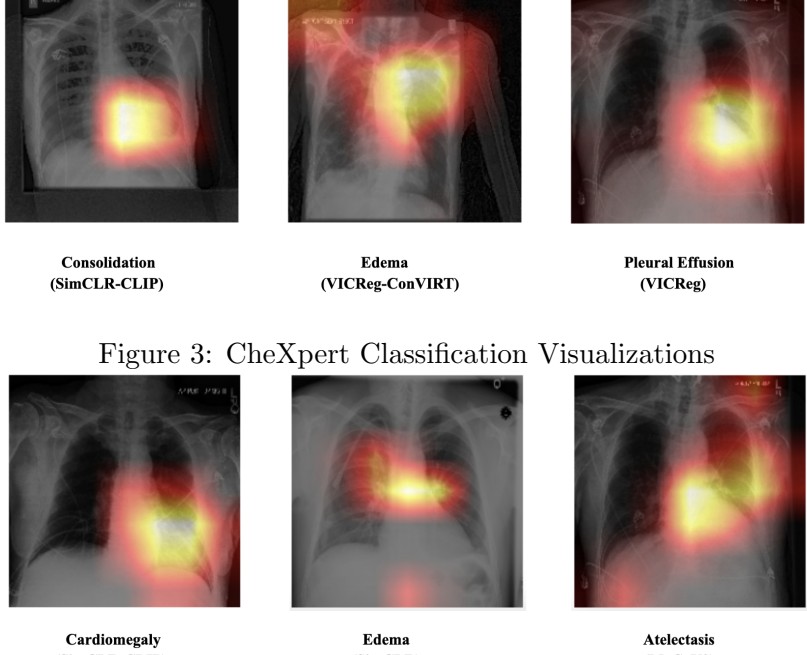

Figure 3: CheXpert Classification Visualizations

Figure 4: NIH-ChestXray-14 Classification Visualization

## Appendix D. Performance on Different Pathology

This section shows per pathology performance on the two datasets with different methods. The AUROC score here is picked by taking mean around each pathology across seeds.

**D.1.** 100% **label fraction**

| Pathologies | CheXpert (AUROC) | | | | |
|---|---|---|---|---|---|
| 100% fine-tuning | ImageNet | CLIP | SimCLR | SimCLR-CLIP | Prevalence (%) |
| No Finding | **90.07** | 87.52 | 86.64 | 88.13 | 7.41 |
| Enlarged Cardiomediastinum | **64.72** | 52.21 | 55.14 | 50.72 | 8.55 |
| **Cardiomegaly** | **86.10** | 81.80 | 82.90 | 82.16 | 13.20 |
| Lung Lesion | 32.80 | 66.66 | 86.06 | **96.01** | 3.53 |
| Lung Opacity | 89.01 | 89.49 | 91.65 | **91.66** | 43.25 |
| **Edema** | 93.91 | 91.99 | 93.16 | **94.97** | 26.96 |
| **Consolidation** | 88.53 | 87.59 | **89.44** | 88.75 | 16.36 |
| Pneumonia | **88.59** | 82.53 | 72.93 | 74.42 | 9.02 |
| **Atelectasis** | 81.07 | 79.16 | 83.61 | **85.07** | 26.17 |
| Pneumothorax | **91.64** | 85.05 | 86.73 | 78.38 | 8.91 |
| **Pleural Effusion** | 93.23 | 92.73 | **93.80** | 93.08 | 37.94 |
| Pleural Other | 95.02 | 92.03 | 80.09 | **97.01** | 1.88 |
| Fracture | 93.51 | 92.76 | 94.90 | **96.46** | 3.46 |

Table 2: Per Pathology Performance for different pre-training methods after 100% fine-tuning

| Pathologies | NIH-ChestX-Ray14 (AUROC) | | | | |
|---|---|---|---|---|---|
| 100% fine-tuning | ImageNet | CLIP | SimCLR | SimCLR-CLIP | Prevalence (%) |
| **Atelectasis** | 80.92 | 82.10 | 80.05 | **82.19** | 10.79 |
| **Cardiomegaly** | 89.89 | 90.71 | 90.05 | **90.75** | 2.59 |
| **Effusion** | 87.25 | 87.36 | 87.35 | **87.45** | 12.28 |
| Infiltration | 69.90 | 70.50 | 70.11 | **70.58** | 17.56 |
| Mass | 82.97 | 83.39 | 82.53 | **84.32** | 5.05 |
| Nodule | 76.88 | 77.17 | 77.19 | **77.33** | 5.95 |
| Pneumonia | 74.35 | 76.14 | 74.81 | **76.33** | 1.08 |
| Pneumothorax | 85.05 | 86.86 | 86.16 | **87.44** | 4.85 |
| **Consolidation** | 79.74 | 80.63 | 79.61 | **80.80** | 4.27 |
| **Edema** | 88.99 | 89.47 | 88.06 | **89.56** | 1.84 |
| Emphysema | 91.91 | 92.57 | 91.63 | **92.69** | 2.27 |
| Fibrosis | 82.70 | 81.64 | 81.09 | **83.07** | 1.61 |
| Pleural Thickening | 77.12 | 78.62 | 76.50 | **77.71** | 3.27 |
| Hernia | 93.20 | 93.81 | 91.54 | **95.08** | 0.19 |

Table 3: Per Pathology Performance for different pre-training methods after 100% fine-tuning

**D.2.** 10% **label fraction**

| Pathologies | CheXpert (AUROC) | | | | |
|---|---|---|---|---|---|
| 10% fine-tuning | ImageNet | CLIP | SimCLR | SimCLR-CLIP | Prevalence |
| No Finding | 85.51 | 90.01 | 89.44 | **90.10** | 7.41 |
| Enlarged Cardiomediastinum | 57.46 | **67.77** | 57.66 | 67.72 | 8.55 |
| Cardiomegaly | 73.67 | 85.20 | 77.76 | **86.16** | 13.20 |
| Lung Opacity | 86.23 | 90.79 | 87.99 | **91.50** | 3.53 |
| Lung Lesion | 48.76 | **56.21** | 46.26 | 55.72 | 43.25 |
| Edema | 90.65 | 93.13 | 89.76 | **93.49** | 26.96 |
| Consolidation | 82.57 | 87.83 | 75.45 | **89.39** | 16.36 |
| Pneumonia | 71.65 | **82.73** | 81.82 | 78.67 | 9.02 |
| Atelectasis | 76.03 | 85.56 | 77.72 | **85.92** | 26.17 |
| Pneumothorax | 70.55 | **89.96** | 74.65 | 89.45 | 8.91 |
| Pleural Effusion | 84.23 | 92.50 | 90.37 | **92.75** | 37.94 |
| Pleural Other | 86.57 | 96.01 | **99.50** | 97.01 | 1.88 |
| Fracture | 81.91 | **91.65** | 82.63 | 90.64 | 3.46 |

Table 4: Per Pathology Performance for different pre-training methods after 10% fine-tuning

| Pathologies | NIH-ChestX-Ray14 (AUROC) | | | | |
|---|---|---|---|---|---|
| 10% fine-tuning | ImageNet | CLIP | SimCLR | SimCLR-CLIP | Prevalence |
| Atelectasis | 76.68 | 78.46 | 75.68 | **78.69** | 10.79 |
| Cardiomegaly | 83.07 | **89.36** | 85.40 | 88.14 | 2.59 |
| Effusion | 85.33 | **87.18** | 85.75 | 86.42 | 12.28 |
| Infiltration | 66.70 | **68.16** | 65.13 | 67.79 | 17.56 |
| Mass | 76.13 | **81.93** | 75.62 | 80.47 | 5.05 |
| Nodule | 68.35 | **73.12** | 65.65 | 72.57 | 5.95 |
| Pneumonia | 70.35 | **73.13** | 68.73 | 70.96 | 1.08 |
| Pneumothorax | 81.37 | **84.77** | 80.61 | 81.90 | 4.85 |
| Consolidation | 78.09 | **78.58** | 76.44 | 78.56 | 4.27 |
| Edema | 86.19 | **86.85** | 86.28 | 86.76 | 1.84 |
| Emphysema | **87.20** | 86.30 | 78.08 | 87.18 | 2.27 |
| Fibrosis | 75.39 | **77.55** | 73.69 | 75.56 | 1.61 |
| Pleural Thickening | 72.89 | 73.09 | 73.69 | **75.34** | 3.27 |
| Hernia | 81.85 | **81.90** | 78.89 | 77.94 | 0.19 |

Table 5: Per Pathology Performance for different pre-training methods after 10% fine-tuning

**D.3.** 1% **label fraction**

| Pathologies | CheXpert (AUROC) | | | | |
|---|---|---|---|---|---|
| 1% fine-tuning | ImageNet | CLIP | SimCLR | SimCLR-CLIP | Prevalence |
| No Finding | 82.65 | **93.39** | 93.00 | 91.63 | 7.41 |
| Enlarged Cardiomediastinum | 48.95 | 51.00 | **69.63** | 46.29 | 8.55 |
| Cardiomegaly | 70.35 | 74.45 | 77.94 | **79.28** | 13.20 |
| Lung Opacity | 72.70 | **90.57** | 83.64 | 86.05 | 3.53 |
| Lung Lesion | 40.30 | 28.35 | 32.83 | **49.25** | 43.25 |
| Edema | 82.54 | **89.68** | 81.26 | 89.47 | 26.96 |
| Consolidation | 82.76 | 84.70 | 85.47 | **86.63** | 16.36 |
| Pneumonia | 59.99 | 79.12 | 73.00 | **83.24** | 9.02 |
| Atelectasis | 75.43 | **83.47** | 80.30 | 78.47 | 26.17 |
| Pneumothorax | 80.88 | **83.07** | 67.17 | 67.54 | 8.91 |
| Pleural Effusion | 78.53 | 87.74 | 82.30 | **91.50** | 37.94 |
| Pleural Other | 78.11 | 95.52 | 68.65 | **95.52** | 1.88 |
| Fracture | 75.20 | **89.77** | 65.76 | 88.66 | 3.46 |

Table 6: Per Pathology Performance for different pre-training methods after 1% fine-tuning

| Pathologies | NIH-ChestX-Ray14 (AUROC) | | | | |
|---|---|---|---|---|---|
| 1% fine-tuning | ImageNet | CLIP | SimCLR | SimCLR-CLIP | Prevalence |
| Atelectasis | 65.17 | 76.14 | 74.18 | **76.25** | 10.79 |
| Cardiomegaly | 53.90 | 68.52 | **68.87** | 65.49 | 2.59 |
| Effusion | 80.12 | **84.87** | 81.45 | 84.37 | 12.28 |
| Infiltration | 63.23 | **66.95** | 64.65 | 65.66 | 17.56 |
| Mass | 67.69 | **79.79** | 63.77 | 74.39 | 5.05 |
| Nodule | 56.40 | 69.54 | 59.79 | **64.61** | 5.95 |
| Pneumonia | 62.14 | **65.86** | 62.39 | 61.75 | 1.08 |
| Pneumothorax | 77.00 | **82.91** | 76.47 | 78.86 | 4.85 |
| Consolidation | 75.38 | **78.33** | 75.09 | 75.93 | 4.27 |
| Edema | 80.58 | 79.52 | **84.38** | 82.29 | 1.84 |
| Emphysema | 73.18 | **85.38** | 72.01 | 68.55 | 2.27 |
| Fibrosis | 69.56 | **72.83** | 64.23 | 68.22 | 1.61 |
| Pleural Thickening | 67.80 | **71.60** | 64.37 | 70.94 | 3.27 |
| Hernia | 52.92 | 68.31 | 60.08 | **79.13** | 0.19 |

Table 7: Per Pathology Performance for different pre-training methods after 1% fine-tuning

## Appendix E. SimCLR and CLIP Performance Trained with Larger Batch Size

We further show performance evaluation on SimCLR and CLIP trained with 1024 and 2048 batch size respectively. Because NVIDIA Tesla V100 32GB GPU does not fit such large batch size, we conducted these experiments on four NVIDIA Tesla A100 GPUs 80GB with rest of the settings to be the same as our previous experiments.

| Methods | CheXpert (AUROC) | | | |
|---|---|---|---|---|
| | LP | FT (1%) | FT (10%) | FT (100%) |
| Random Init. | $62.91 \pm 0.62$ | $62.88 \pm 1.13$ | $70.36 \pm 0.98$ | $78.28 \pm 1.10$ |
| ImageNet Init. | $67.12 \pm 0.24$ | $69.33 \pm 0.15$ | $77.98 \pm 0.28$ | $82.29 \pm 2.79$ |
| CLIP | $\mathbf{78.13} \pm 0.24$ | $\mathbf{78.38} \pm 1.60$ | $\mathbf{79.75} \pm 2.36$ | $\mathbf{85.91} \pm 1.39$ |
| SimCLR | $74.83 \pm 0.06$ | $76.17 \pm 2.03$ | $79.09 \pm 1.67$ | $84.63 \pm 0.70$ |
| Methods | NIH-ChestX-Ray14 (AUROC) | | | |
| | LP | FT (1%) | FT (10%) | FT (100%) |
| Random Init. | $62.54 \pm 0.34$ | $58.30 \pm 2.06$ | $70.06 \pm 1.02$ | $78.52 \pm 0.17$ |
| ImageNet Init. | $75.46 \pm 0.78$ | $67.18 \pm 0.07$ | $76.44 \pm 0.63$ | $82.70 \pm 0.23$ |
| CLIP | $\mathbf{83.08} \pm 0.07$ | $\mathbf{70.06} \pm 0.84$ | $\mathbf{77.64} \pm 0.37$ | $\mathbf{82.88} \pm 0.14$ |
| SimCLR | $77.84 \pm 0.10$ | $68.26 \pm 0.41$ | $77.07 \pm 0.23$ | $82.88 \pm 0.21$ |

(a) — rows: Random Init., ImageNet Init., CLIP, SimCLR (CheXpert)

(b) — rows: Random Init., ImageNet Init., CLIP, SimCLR (NIH-ChestX-Ray14)

Table 8: Downstream results for CLIP and SimCLR with 1024 batch size. The experiment in this section is exactly following Table 1

| Methods | CheXpert (AUROC) | | | |
|---|---|---|---|---|
| | LP | FT (1%) | FT (10%) | FT (100%) |
| Random Init. | $62.91 \pm 0.62$ | $62.88 \pm 1.13$ | $70.36 \pm 0.98$ | $78.28 \pm 1.10$ |
| ImageNet Init. | $67.12 \pm 0.24$ | $69.33 \pm 0.15$ | $77.98 \pm 0.28$ | $82.29 \pm 2.79$ |
| CLIP | $\mathbf{77.86} \pm 0.29$ | $\mathbf{79.00} \pm 2.46$ | $\mathbf{81.67} \pm 0.14$ | $\mathbf{84.65} \pm 0.29$ |
| SimCLR | $74.91 \pm 0.37$ | $77.67 \pm 1.71$ | $79.31 \pm 1.37$ | $83.04 \pm 0.66$ |
| Methods | NIH-ChestX-Ray14 (AUROC) | | | |
| | LP | FT (1%) | FT (10%) | FT (100%) |
| Random Init. | $62.54 \pm 0.34$ | $58.30 \pm 2.06$ | $70.06 \pm 1.02$ | $78.52 \pm 0.17$ |
| ImageNet Init. | $75.46 \pm 0.78$ | $67.18 \pm 0.07$ | $76.44 \pm 0.63$ | $82.70 \pm 0.23$ |
| CLIP | $\mathbf{82.60} \pm 0.18$ | $\mathbf{73.20} \pm 1.14$ | $\mathbf{79.62} \pm 0.44$ | $\mathbf{83.56} \pm 0.29$ |
| SimCLR | $77.32 \pm 0.11$ | $69.18 \pm 0.83$ | $78.49 \pm 0.52$ | $83.45 \pm 0.40$ |

(a) — rows: Random Init., ImageNet Init., CLIP, SimCLR (CheXpert)

(b) — rows: Random Init., ImageNet Init., CLIP, SimCLR (NIH-ChestX-Ray14)

Table 9: Downstream results for CLIP and SimCLR with 2048 batch size. The experiment in this section is exactly following Table 1

## Appendix F. Additional Visualization for t-SNE

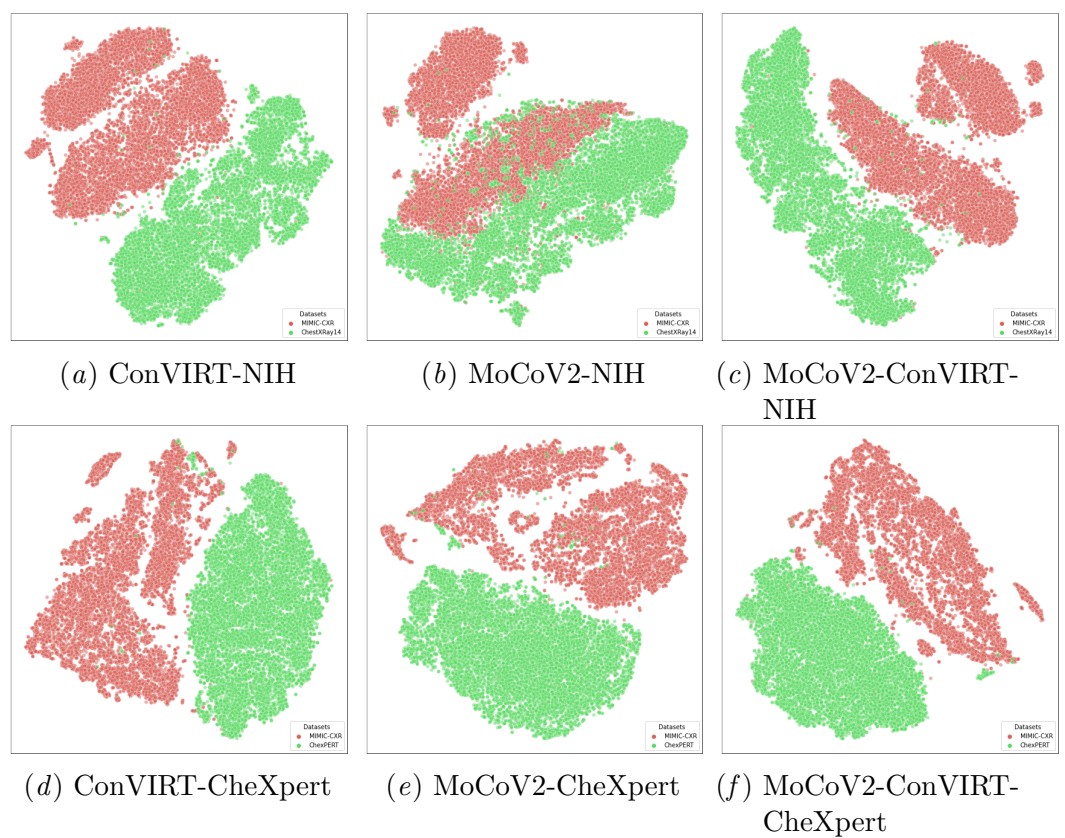

$(a)$ ConVIRT-NIH  $\qquad$ $(b)$ MoCoV2-NIH  $\qquad$ $(c)$ MoCoV2-ConVIRT-NIH

$(d)$ ConVIRT-CheXpert  $\qquad$ $(e)$ MoCoV2-CheXpert  $\qquad$ $(f)$ MoCoV2-ConVIRT-CheXpert

Figure 5: Figures for t-SNE embedding with LPFT downstream training strategies for Con-VIRT, MoCoV2, MoCoV2-ConVIRT pretrained backbone

