# OpenReview forum: "Radiology Reports Improve Visual Representations Learned from Radiographs"
_MIDL.io/2023/Conference — MIDL 2023 Poster_

### Official Review · Reviewer_ytN5 · 2023-01-29

**Confidence:** 5
**Preliminary Rating:** 2

**Summary:**

The key idea of this research paper is to use multi-modal data, or data from multiple sources, to build a framework that can be used to solve tasks in the biomedical domain. The researchers conducted experiments comparing self-supervised learning and joint learning with multi-modal and self-supervised models on limited labelled datasets with 1% and 10% labelled data. They found that joint learning outperformed self-supervised learning alone, while being on par with multi-modal learning. Additionally, they found that multi modal was more robust across different environments than other methods tested.

**Strengths:**

The strengths of this paper are that it provides a novel approach to solving tasks by leveraging multi-modal data. The experiments conducted were thorough and well designed, providing evidence for their conclusions. Additionally, the code from the research project is freely available online which allows other researchers to build upon their work or replicate it if needed. CIs have been provided for the evaluation.

**Weaknesses:**

- the authors basically compare a setup with partly enriched labels to self-supervised learning from images only.
- what is meant by visual representation? t-SNE?
- the used datasets are multi-label, multi-class? How was this taken into account when computing the AUROC? What is meant by "mean AUROC from all pathology"?
- how does OOD play a role here? There are no known or unknowns and everything is fine-tuned on comprehensive label data, neither prediction confidence is evaluated.

**Deanonymize Review:**

no

**Detailed Comments:**

I found no typos but the paper is difficult to follow and it is unclear what the main motivation of this paper actually is.
There is a lot of related work regarding multi-modal learning missing, e.g., automated report generation and recently image from text generation (google scholar will reveal a lot of references regarding multi-modal learning with CXR images; I don't provide any here to maintain anonymity)


**Paper Type:**

validation/application paper

**Questions To Address In The Rebuttal:**

- what problem is actually addressed?
- clarification on my terminology questions, .e.g., OOD, visual representation etc.
- preformance results re multi-class, multi-label data
- What does "Code will be available in the camera-ready version or upon request." mean?

---

### Official Review · Reviewer_Z5Yw · 2023-02-03

**Confidence:** 3
**Preliminary Rating:** 4
**Recommendation:** Poster

**Summary:**

The authors compare the performances of multi-modal learning, self-supervised learning and joint learning with both multi-modality and self-supervision. These models are initially trained with the MIMIC-CXR data. The models are then fine-tuned only at the final layer or all the layers with different amounts of labelled data from the CheXPert and NIH Chest XRay-14 data sets and their accuracies are compared. On fine-tuning with 1% labelled data, the multi-modal and joint learning models have a better performance than models trained with only self-supervision. Further, the feature representations of the different models are visualised as a t-SNE plot.

**Strengths:**

1. The authors have performed an exhaustive comparation of the different self-supervised learning approaches, multi-modal learning approaches and their combinations.
2. When trained with only 1% labelled data of CheXPert data which is an external data set, the multimodal and joint learning approaches perform better than the only self-supervised learning approach.

**Weaknesses:**

1. With 1% or 10% labelled data, there are performance improvements of multi-modal learning over self-supervised learning. In case of NIH ChestXRay data, the improvements are only marginal. However, this is not a surprising result since multi-modal learning benefits from additional text data which is not available to the self-supervised learning approach.  In my opinion, it is not a fair comparison of the models since these models are meant for different learning scenarios.
2.It is hard to make any conclusions from the t-SNE plots shown in Figure 2. One would expect the t-SNE plots to show more class-separations for each data set. Although there are only complete clusters for a data set.
3. The authors claim that multi-modality improves performance. However, the evaluation results in Table 1. show no significant performance improvements of the self-supervised or multi-modal learning models over ImageNet initialized models when fine-tuned with 100% data. With 100% data, it is likely that the model is fit to the new data and the learning regimes used previously doesn’t matter anymore.

**Deanonymize Review:**

no

**Paper Type:**

validation/application paper

**Questions To Address In The Rebuttal:**

1. What is the test set used for evaluation of the different models?
2. Multimodal models can perform better than self-supervised, but it needs additional text data which may not be available in all cases. On the other hand, self-supervised models do not require any additional text or labels while showing only a smaller performance drop. Please comment.
3. While for CheXPert in 1% settings multi-modality or joint learning shows significant improvement, the same is not observed for NIH ChestXRay 14 data. Could it be possible that the CheXPert data is closer to the MIMIC-CXR data distribution?
4. After full fine tuning with a different data set, the models might have fit to the new data set and “forgotten” the trends learnt from the original MIMIC-CXR data. How does greater separation of different data sets after fine-tuning with 100% OOD data support the claim that multi-modal and joint learning approaches are better than only self-supervised learning approaches?

---

### Official Review · Reviewer_f4WF · 2023-02-10

**Confidence:** 3
**Preliminary Rating:** 4
**Recommendation:** Poster

**Summary:**

This paper concerns the question: For multi-modal learning, self-supervised learning and joint learning using both learning strategies, which one improves the visual representation for downstream chest radiographs classification tasks the most? And the conclusion is multi-modal learning and joint training performs better than self-supervised learning with limited supervised data, but the performance difference between all methods is largely decreased when data size scales up.

**Strengths:**

This paper leverages CLIP, ConVIRT, MocoV2, SimCLR and VICReg method in the experiment. CLIP and ConVIRT are contrastive multi-modal learning methods, and the others are self-supervised learning methods. I am happy to see so many methods participating in the comparison. Finding the strengths of self-supervised learning method and contrastive multi-modal learning method may provide some inspiration for follow-up work.

**Weaknesses:**

I have not seen your own pipeline or framework proposed in the paper. For example, Figure 1 is just a simple representation of the experimental workflow for other methods. In the Method section, only the loss functions of other methods are introduced.
Besides, the experimental results of the two methods on different proportions of data sets seem to be able to give further explanations, and some examples can be listed to show the classification results. From the point of view of the whole paper, this is a comparative experiment, which innovation in the model structure is not so sufficient but draws an interesting conclusion. Considering the massive image data of the existing CXR dataset, the classification results obtained by using these two methods are very different, so the reference significance on large datasets is open to question.

**Deanonymize Review:**

no

**Paper Type:**

validation/application paper

**Questions To Address In The Rebuttal:**

1. A comparison of the classification results of some images can be listed.
2. The difference of experimental results of two kinds of methods may need an explanation.
3. The "CheXPert" may be "CheXpert".

---

### Meta-Review · Area_Chair_e84c · 2023-02-25

**Recommendation:** Accept (Poster)
**Confidence:** 3

**Metareview:**

This paper explores designing an efficient framework that adapts to multiple modalities for biomedical data and found that joint learning with multi-modal and self-supervised models outperforms self-supervised learning and is at par with multi-modal learning, and multi-modal learning is generally more robust on out-of-distribution datasets.
Two reviewers accept this paper while one rejects it.
The authors have robustly responded to the comments and the responses solve most of the issues raised by reviewers.
recommend: accept